# Molecular Detection of *Leishmania* (*V.*) *braziliensis* and *Leishmania* (*M.*) *martiniquensis* Infecting Domestic Animals from Panama, Central America

**DOI:** 10.3390/ani15182677

**Published:** 2025-09-12

**Authors:** Vanessa Pineda, Jose E. Calzada, Santiago Montilla, Indra Rodríguez, Erika Howard, Alicia I. Torres, Vanessa Vasquez, Adelys Reina, Azael Saldaña, Kadir González

**Affiliations:** 1Departamento de Investigación en Parasitología, Instituto Conmemorativo Gorgas de Estudios de la Salud, Panama 0816-02593, Panama; vpineda@gorgas.gob.pa (V.P.); jcalzada@gorgas.gob.pa (J.E.C.); vvasquez@gorgas.gob.pa (V.V.); areina@gorgas.gob.pa (A.R.); 2Facultad de Medicina Veterinaria, Universidad de Panama, Panama 0816-03366, Panama; 3Estación Biomédica Experimental, Instituto Conmemorativo Gorgas de Estudios de la Salud, Panama 0816-02593, Panama; smontilla@gorgas.gob.pa; 4Clínica Veterinaria Mundo Animal, Panama 0816-03352, Panama; indrarl24@gmail.com; 5Clínica de Diagnóstico Integral Veterinario (CADIV), Panama 0816-46686, Panama; howarderika02@gmail.com; 6Departamento de Clínicas y Cirugías Veterinarias, Facultad de Medicina Veterinaria, Complejo Hospitalario Veterinario de Corozal, Universidad de Panamá, Panama 0816-03366, Panama; alicia.torresm@up.ac.pa; 7Centro de Investigación y Diagnóstico de Enfermedades Parasitarias (CIDEP), Facultad de Medicina, Universidad de Panamá, Panama 0816-03366, Panama; 8Departamento de Microbiología Humana, Facultad de Medicina, Universidad de Panamá, Panama 0816-03366, Panama

**Keywords:** *Leishmania braziliensis*, *Leishmania martiniquensis*, dog, horse, leishmaniasis, Panama

## Abstract

*Cutaneous leishmaniasis* is a disease transmitted by sandflies that mainly affects the skin. It is quite common in humans from Panama, but there has been little research on its presence in domestic animals. In this study, we evaluated twelve domestic animals with suspected leishmaniasis lesions between 2021 and 2025. Six of them (50%) tested positive: five dogs and one horse. Remarkably, three of the dogs were infected with *Leishmania* (*V.*) *braziliensis*, marking the first molecularly confirmed cases in dogs in Panama and Central America. Two other dogs had *Leishmania* (*L.*) *infantum*, but these cases were imported from other countries. The horse was infected with *Leishmania* (*M.*) *martiniquensis*, a species not previously reported in horses from this region. These findings suggest that more types of *Leishmania* can affect animals in Panama than we previously realized. This is especially important because animals might play a role in spreading the disease. So, it is essential to keep an eye on both people and animals, particularly in rural areas, and to consider leishmaniasis if pets develop skin wounds. We also need more research to understand how the disease spreads and how we can protect the health of both humans and animals in the future.

## 1. Introduction

*American cutaneous leishmaniasis* (ACL) is a common but neglected tropical zoonotic disease in Central America. Mammalian reservoirs play a crucial role in its epidemiology by serving as infection sources for sandflies, which act as vectors transmitting *Leishmania* spp. to humans [1,2]. In this region, four species within the subgenus *Viannia* (*L.* (*V.*) *braziliensis*, *L.* (*V.*) *panamensis*, *L.* (*V.*) *guyanensis*, and *L.* (*V.*) *naiffi*) and one within the subgenus *Leishmania* (*L.* (*L.*) *mexicana*) have been identified as etiological agents of ACL in humans [3,4].

In Panama, the southernmost country of Central America, ACL is endemic in both its cutaneous and mucosal clinical forms. Recent epidemiological reports place Panama among the countries in the Americas with the highest incidence and transmission risk of ACL [3]. *Leishmania* (*V.*) *panamensis* is the predominant etiological agent of human ACL in the country, responsible for over 95% of reported cases [5,6,7,8]. However, the recent implementation of molecular techniques has led to the sporadic detection of other *Leishmania* species in human cutaneous lesions, indicating their circulation in endemic areas, albeit at low frequencies [5,6,9].

Canine and equine leishmaniasis have been reported throughout the Americas, involving different *Leishmania* subgenera (*Viannia*, *Leishmania*, and *Mundinia*) [10,11,12,13,14,15,16]. However, in contrast to the well-studied epidemiology of human ACL, information on *Leishmania* species infecting domestic animals in Panama remains limited. Dogs and horses, frequently found in endemic rural and deforested areas, may play a role in parasite transmission due to their proximity to humans and vector exposure. Environmental changes such as deforestation and land use shifts can alter vector dynamics, affect parasite reservoirs, and increase human risk. Identifying *Leishmania* species in domestic animals is therefore key for understanding their role in maintaining transmission cycles and for assessing zoonotic risks [17,18]. This study presents the first molecular evidence of *L.* (*V.*) *braziliensis* and *L.* (*M.*) *martiniquensis* infections in naturally infected local dogs and a horse with cutaneous lesions from endemic areas in Panama.

## 2. Materials and Methods

### 2.1. Study Design

Samples from domestic animals (blood, serum, and/or biopsies) with a clinical suspicion of leishmaniasis were analyzed at the Instituto Conmemorativo Gorgas de Estudios de la Salud (ICGES) from 2021 to 2025. These samples were referred to the ICGES by private veterinary clinics for confirmation of the initial diagnosis. The ICGES is the Reference Center for leishmaniasis diagnosis in Panama. The suspected samples originated from ten dogs (*Canis lupus familiaris*) and two horses (*Equus ferus caballus*) from endemic regions of cutaneous leishmaniasis (CL) in the country, within the areas of Panama (Panama City), Panama Oeste (La Chorrera), Panama Este (Chepo), Panama Norte (Chilibre), and Colon (Escobal). Samples were stored at −20 °C until processed, depending on the type of sample.

### 2.2. Serological Analysis

Serum samples were analyzed for the detection of *Leishmania* antibodies using two serological methods. The indirect immunofluorescence assay (IFA) detects anti-*Leishmania* antibodies by incubating serum samples with antigen-coated slides, followed by binding of a fluorescent secondary antibody and microscopic visualization. The antigen was obtained from *L.* (*V.*) *panamensis* promastigotes, and we used an anti-Dog IgG (whole molecule)—FITC antibody produced in rabbit (Sigma: F7884, Saint Louis, MO, USA). A titer of ≥1:64 was considered positive, based on prior validation studies in dogs from Panama [19]. The second method was a rapid immunochromatographic test (DPP^®^, Bio Manguinhos, Rio de Janeiro, Brazil), which detects antibodies against the rK39 recombinant antigen using lateral flow technology and is commonly used for the diagnosis of canine visceral leishmaniasis (CVL). Results were interpreted visually according to the manufacturer’s instructions.

### 2.3. Molecular Analysis

#### 2.3.1. DNA Extraction

DNA was extracted from blood samples using the QIAamp DNA Blood Mini Kit (Qiagen, Redwood City, CA, USA). For biopsies, the Wizard™ Genomic DNA Purification Kit (Promega: A1120, Madison, WI, USA) was used, following the manufacturer’s instructions.

#### 2.3.2. PCR

Four conventional PCR tests were performed for the detection of *Leishmania* (Table 1): (a) *kDNA Viannia* genus-specific PCR: targeting the kinetoplast minicircle region for the detection of *Leishmania Viannia* subgenus including *L. braziliensis* complex [20]; (b) *Hsp-70* gene PCR: used for the detection and characterization of *Leishmania Viannia* and *Mundinia* subgenus [21]; (c) L150/L151-PCR: for *kDNA minicircle gene* specific to the *Leishmania* genus [22]; and (d) *internal transcribed spacer 1* (*ITS1*) region for the *rRNA* gene for the detection of *L.* (*M.*) *martiniquensis* [23,24,25,26].

### 2.4. Sequencing

*Hsp70*-PCR products were sequenced using the Sanger technique. In the case of the horse, amplification and Sanger sequencing of the *ITS1* region of the *rRNA* gene (379 bp product) were performed to confirm *L.* (*M.*) *martiniquensis* infection. Although the methodology referenced [25] was originally developed for Old World *Leishmania* species, it was suitable for *L.* (*M.*) *martiniquensis* due to its phylogenetic placement within the *Mundinia* subgenus, which clusters near Old World species [23,24,26]. *ITS1* sequences were analyzed by BLAST v2.15.0 and phylogenetic reconstruction to establish species identity. A second molecular approach (*Hsp-70* gene sequencing) was used to confirm this result [23,24,25,26]. Sanger sequencing reactions were performed using the Applied Biosystems™ BigDye Terminator v3.1 cycling kit (Thermo Fisher Scientific: 4337454, Vilnius, Lithuania). Reaction products were purified with the BigDye XTerminator kit (Applied Biosystems, Waltham, MA, USA) and analyzed using the 3130xl Genetic Analyzer sequencer (Applied Biosystems, Foster, CA, USA). Resulting sequences were edited using Sequencher software v4.6 and their homology with those available in GenBank database was assessed by performing a BLAST search from the National Center for Biotechnology Information Database (http://www.ncbi.nlm.nih.gov/BLAST/, accessed on 22 July 2025). Sequence alignment and phylogenetic analyses were performed as described [5]. A consensus tree was summarized from sampled trees and visualized with FigTree v.1.4.4. To create the phylogenetic tree of the *Hsp-70* gene with the sequences of *L.* (*V.*) *braziliensis* and *L.* (*M.*) *martiniquensis*, the best-fit model TN+F+I and an ultra-fast bootstrap with 1000 replicates were used. In the case of the *ITS1* gene with the sequence of *L.* (*M.*) *martiniquensis*, the best-fit model HKY+F+I and an ultra-fast bootstrap with 1000 replicates were employed. Representative sequences obtained from NCBI GenBank of *Leishmania* and *Endotrypanum* were used for phylogenetic analysis with the *Hsp-70* gene, and existing sequences of *Leishmania* were used for phylogenetic analysis with the *ITS1* gene. Nucleotide sequences are available in NCBI GenBank under the accession numbers PV658270 (ITS1) and PV844900-PV844903 (Hsp70).

### 2.5. Ethical Statement

ICGES is the national reference laboratory for *Leishmania* diagnosis. As leishmaniasis is a notifiable disease in Panama, samples from this study were analyzed as part of the surveillance activities approved in the National Commission for the Control and Prevention of Tropical Neglected Diseases. The use of animal biological samples for this study was granted exemption (008/CIUCAL-ICGES25) from the Comité Institucional para el Uso y Cuidado de Animales de Laboratorio (CIUCAL-ICGES). Samples were coded and anonymized to protect the confidentiality of the study subjects and owners.

## 3. Results

Between 2021 and 2025, twelve domestic animals, including ten dogs and two horses from endemic areas of CL, were tested for clinical suspicion of leishmaniasis. *Leishmania* infection was confirmed by serological testing and/or conventional PCR in 50% of cases (6/12), with one horse and five dogs identified (Table 2). Of the canine cases, three were classified as CL and two as CVL, while the horse case was diagnosed with CL.

### 3.1. Canine Cases

Of the 10 canine cases analyzed, three local dogs were diagnosed with CL (CF-W01, CF-P02, CF-L03) by clinical and laboratory tests. All three dogs had ulcerated lesions on their noses and had no travel history to another country (Appendix A). The remaining dogs were tested for clinical suspicion of CVL (7/10), with two testing positive for *Leishmania* (CF-J07, CF-A09) via serological (DPP rk39 and IFA) and molecular (*kDNA*, *Hsp-70*) tests. One CVL-positive dog had a travel history to Brazil and exhibited overt clinical signs compatible with leishmaniasis, including dermatological alterations. The second dog diagnosed with CVL had traveled to Spain and presented an enlarged spleen as the only clinical sign consistent with the infection. The remaining five dogs with suspected CVL presented one or more of the following non-specific clinical signs: weight loss, lethargy, enlarged lymph nodes, and different types of skin lesions. *Leishmania* infections in the three CL cases were characterized as *L.* (*V.*) *braziliensis* by *Hsp70*-RFLP and Sanger sequencing (Figure 1). In the two CVL cases, *Hsp70*-RFLP and sequencing were not possible due to insufficient amplified DNA.

### 3.2. Equine Cases

Of the two horses tested for clinical suspicion of CL, one tested positive for *Leishmania* (EC-H01) using molecular methodologies involving the *Hsp-70* and *kDNA* genes. This sample was further characterized as *L.* (*M.*) *martiniquensis* by sequencing the *Hsp-70* and *ITS1* genes (Figure 1). The horse exhibited nodular and ulcerated lesions on the ears, originated from a CL endemic area, and had no history of travel outside the country. The CL-negative horse had an ulcerated lesion with crust, purulent discharge, and intense itching. This horse was diagnosed with cutaneous habronemiasis, a parasitic disease of horses frequently prevalent in tropical climates caused by *Habronema* larvae transmitted by flies (*Musca domestica* and *Stomoxys calcitrans*), which cause skin, eye, or gastric lesions. Gross lesions are manifest as granulomatous ulcers that can be confused with ulcerated lesions caused by *Leishmania* parasites [27,28].

## 4. Discussion

In the Americas, wild animal reservoirs play a crucial role in maintaining the zoonotic transmission cycle of *Leishmania* species indigenous to the Neotropical region [1,2,3]. The exception is *L.* (*L.*) *infantum*, introduced to the continent during the colonial era [29], which is primarily maintained in a zoonotic transmission cycle where dogs serve as the main reservoir hosts [10]. In Panama, *L.* (*L.*) *infantum* has only been recently detected in dogs imported from the Mediterranean Basin and South America [30]. In this study, we report a case of *L.* (*L.*) *infantum* infection in a local dog from a recognized ACL-endemic region of the country. However, further investigation of its travel history revealed that although the dog was born in Panama, it lived for a short time in Zaragoza, Spain before returning, confirming it as a new imported case.

Although domestic animals are frequently exposed to *Leishmania Viannia* species in endemic areas of Panama [19], their role as reservoir hosts in the zoonotic transmission of ACL remains unclear. Canine CL cases were documented in the 1970s in rural communities of central Panama [31]. Although the infecting *Leishmania* species were not identified using laboratory methods, the authors hypothesized *L.* (*V.*) *braziliensis* as the etiological agent based on ecological and epidemiological evidence. Using molecular techniques, here we detected *L.* (*V.*) *braziliensis* infections in three local dogs from recognized ACL-endemic areas of Panama, where *L.* (*V.*) *panamensis* is the predominant species. These three dogs presented cutaneous and mucosal ulcers consistent with *L.* (*V.*) *braziliensis* infections [32] (Appendix A). While *L.* (*V.*) *braziliensis* infections in humans have been reported at low frequencies in Panama [5], these findings represent the first molecularly documented cases of *L.* (*V.*) *braziliensis* in dogs from the country and, more broadly, from any Central American nation where human infections by this species have been recorded. In South America, *L.* (*V.*) *braziliensis* is the most common causative agent of ACL in dogs [10,32]. These findings suggest that the co-circulation of *L.* (*V.*) *braziliensis* in ACL-endemic regions of Panama may be more frequent than previously reported. Further research is required to assess the epidemiological significance of *L.* (*V.*) *braziliensis* infections in dogs, including their potential role as competent reservoirs in the transmission cycle.

However, the epidemiological significance of *L.* (*V.*) *braziliensis* infections in dogs in Panama requires cautious interpretation. The small number of cases limits generalization, and the mere presence of infection does not confirm reservoir competence. Factors such as parasitemia levels, duration of infection, and the ability of local vectors to acquire and transmit the parasite from dogs remain to be elucidated.

It is also important to consider the possibility of *L.* (*Viannia*) hybrid strains in Panama. A previous study conducted in central Panama detected molecular signatures suggestive of *L.* (*V.*) *braziliensis*/*L.* (*V.*) *panamensis* hybrids in human cutaneous lesions [5]. Although phylogenetic analysis demonstrated the presence of *L.* (*V.*) *braziliensis*/*L.* (*V.*) *guyanensis* hybrids, we cannot rule out the possibility of the presence of both hybrids in the country. Such hybrids may complicate molecular identification, particularly when using single-locus markers such as *Hsp-70*. Although our *Hsp-70*-based phylogeny grouped the sequences within the *L.* (*V.*) *braziliensis* clade, we cannot rule out the presence of genetic introgression or hybridization with *L.* (*V.*) *panamensis*, especially given the geographic overlap of these species in endemic areas. Future studies using multilocus genotyping or whole-genome sequencing will be essential to clarify this issue.

In the Americas, natural *Leishmania* infections in horses have been frequently documented, with *L.* (*V.*) *braziliensis* and *L.* (*L.*) *infantum* being the most detected species [15,33]. However, in recent years, *L.* (*M.*) *martiniquensis* has emerged as an important species infecting equine [34,35]. Given their frequent exposure to *Leishmania*-infected sandflies in endemic regions and their close interactions with humans and other domestic animals, assessing *Leishmania* infections in horses is essential to elucidate their epidemiological role, zoonotic potential, and clinical significance.

In Panama, several species of *Lutzomyia* sand flies have been implicated in the transmission of *L.* (*V.*) *panamensis* and *L.* (*V.*) *braziliensis*, particularly *Lutzomyia gomezi*, *L. panamensis*, and *L. trapidoi* [8,36,37]. However, the vector responsible for the transmission of *L.* (*M.*) *martiniquensis* remains unknown. In other endemic areas such as Southeast Asia, *Sergentomyia* spp. and *Phlebotomus* spp. have been suggested as potential vectors of *Mundinia* species [34]. Given the presence of *L.* (*M.*) *martiniquensis* in a horse from Panama without a travel history, and its phylogenetic confirmation, it is plausible that local phlebotomine species may be involved in its transmission. Entomological investigations are needed to identify potential vector species and assess their infection rates with *Mundinia* parasites in Central America.

In this study, we identified a horse with cutaneous lesions naturally infected with *L.* (*M.*) *martiniquensis* using two molecular markers (Figure 1). The horse was born and raised in Panama and had no travel history outside the country, providing strong evidence that the infection was locally acquired. This represents the first confirmed report of *L.* (*V.*) *martiniquensis* infecting a horse in Central America. A recent study from the neighboring country of Costa Rica reported five equine cases of leishmaniasis with cutaneous lesions diagnosed through immunohistochemical detection; however, the causative *Leishmania* species in those infections were not identified [38].

One limitation of our study is the use of single-locus markers (*Hsp-70* and *ITS1*) for phylogenetic reconstruction, which may lead to lower bootstrap support in certain branches of the trees, as observed in Figure 1. While these genes are commonly used for *Leishmania* species identification [21,35,39,40], they may not provide sufficient resolution for deeper evolutionary relationships. The use of multilocus or whole-genome sequencing approaches would improve phylogenetic robustness and should be considered in future studies. Nevertheless, the consistent results across two genetic markers and the high sequence identity with reference strains support the accuracy of our species-level identifications. In addition to the molecular findings, the clinical and epidemiological context strengthens our identification. The cutaneous lesion characteristics observed in the horse, along with its origin in a tropical region, are consistent with previously reported cases of leishmaniasis caused by *L.* (*M.*) *martiniquensis* [34,35]. Taken together, the molecular, clinical, and epidemiological evidence strongly supports *L.* (*M.*) *martiniquensis* as the etiological agent in this case.

The detection of *L.* (*M.*) *martiniquensis* in a horse in Panama underscores the need to expand epidemiological surveillance in domestic animals and potential vectors in the region. It is essential to investigate whether its emergence in Panama is linked to environmental factors such as deforestation and climate change, which may be facilitating the expansion of its vectors and reservoirs. Additionally, this finding raises important questions regarding the possible circulation of *L.* (*M.*) *martiniquensis* in humans in Panama, which may have gone undiagnosed due to limitations in conventional detection methods. Enhanced molecular and epidemiological studies are necessary to better understand the distribution, transmission dynamics, and potential public health implications of *L.* (*M.*) *martiniquensis* in the region.

## 5. Conclusions

Our findings provide preliminary molecular evidence of the diversity of *Leishmania* species infecting domestic animals in Panama, including the first molecular documentation of *L.* (*V.*) *braziliensis* in local dogs and *L.* (*M.*) *martiniquensis* in a horse. While these results expand our understanding of potential animal involvement in the transmission cycle of cutaneous leishmaniasis, the analyses were based on two single-locus markers, which generally provided lower phylogenetic resolution. Consequently, species identification should be interpreted with caution. Future studies using multilocus genotyping or genomic approaches are essential to confirm these findings and to explore the possibility of hybrid strains or cryptic diversity. Nonetheless, our results underscore the importance of strengthening surveillance efforts in domestic animals under a One Health framework to better understand the eco-epidemiology of *Leishmania* transmission in Central America.

## Figures and Tables

**Figure 1 animals-15-02677-f001:**
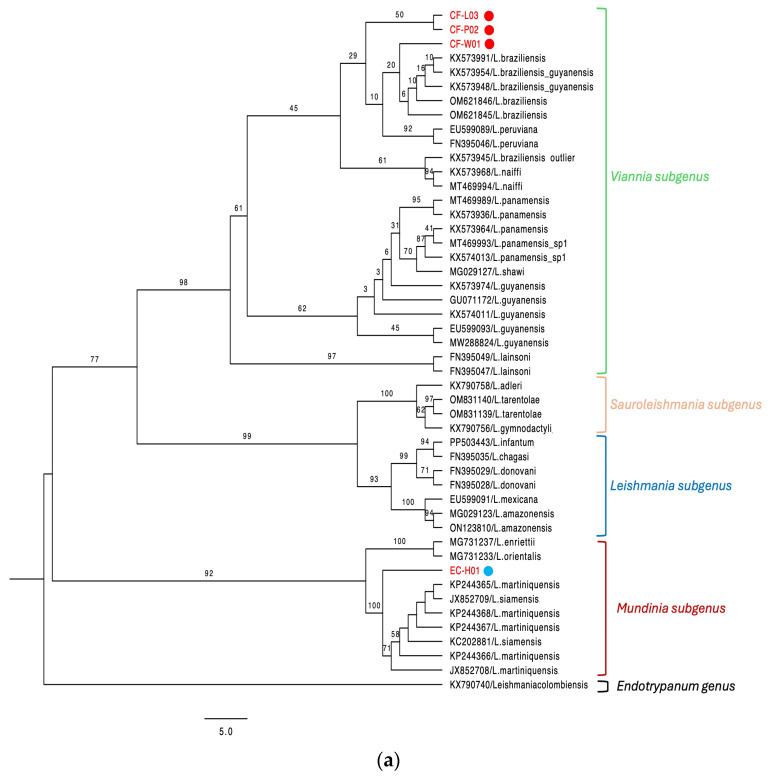
(**a**) Phylogenetic tree with the *Hsp-70* gene sequences of *L.* (*V.*) *braziliensis* (red circle) and *L.* (*M.*) *martiniquensis* (blue circle). The best-fit model TN+F+I and an ultra-fast bootstrap with 1000 replicates were used. (**b**) Phylogenetic tree with the *ITS1* gene sequence of *L.* (*M.*) *martiniquensis* (blue circle) obtained from samples of domestic animals with CL. The best-fit model, HKY+F+I, and an ultra-fast bootstrap with 1000 replicates were used.

**Table 1 animals-15-02677-t001:** PCR protocols used to identify *Leishmania* parasites from domestic animals.

PCR	Primers	Primers Concentrations	DNA Used per Reaction	PCR Conditions	PCR Products Size
*kDNA* *Viannia* Vergel et al., 2005 [20]	B1: 5′-GGGGTTGGTGTAATATAGTGG-3′ LV: 5′-ATTTTTGAACGGGGTTTCTG-3′	0.6 μm/L each one	5 μL	5 cycles of 95 °C × 6 min 95 °C × 30 seg 64.5 °C × 2 min 72 °C × 1 min 35 cycles of 95 °C × 30 seg 64 °C × 1 min 72 °C × 1 min 72 °C × 10 min	750 bp
*Hsp-70* Montalvo et al., 2012 [21]	Hsp70-F25: 5′-GGACGCCGGCACGATTKCT-3′ Hsp70-R1310: 5′-CCTGGTTGTTGTTCAGCCACTC-3′	0.6 μm/L each one	5 μL	94 °C × 5 min 33 cycles of 94 °C × 30 seg 61 °C × 1 min 72 °C × 3 min 72 °C × 10 min	1286 bp
*kDNA* L150/151 Marques et al., 2006 [22]	L-150: 5′-GGG(G/T)AGGGGCGTTCT(G/C)CGAA-3′ L-151: 5′-(G/C)(G/C)(G/C)A/C)CTAT(A/T)TTACACCAACCCC-3′	0.5 μm/L each one	5 μL	94 °C × 4 min 33 cycles of 94 °C × 30 seg 52.3 °C × 30 seg 72 °C × 30 seg 72 °C × 10 min	120 bp
*ITS1* Spanakos et al., 2008 [25]	LeR: 5′-CCAAGTCATCCATCGCGACACG-3′ LeF 5′-TCCGCCCGAAAGTTCACCGATA-3′	1.0 μm/L each one	5 μL	95 °C × 5 min 40 cycles of 95 °C × 1 min 65 °C × 1 min 72 °C × 1 min 72 °C × 10 min	379 bp

**Table 2 animals-15-02677-t002:** Detection of *Leishmania* in domestic animals, including diagnostic results, species identification, and region. N/A: Not available or not applicable.

ID	Animal Species	Sample Type	Region	DPP rk39	IFI	PCR *kDNA* *	PCR *Hsp-70*	*Leishmania* Species	GenBank Accession
CF-W01	Dog	Biopsy	Colon	N/A	N/A	+	+	*L.* (*V.*) *braziliensis*	PV844902
CF-P02	Dog	Biopsy	Colon	N/A	N/A	+	+	*L.* (*V.*) *braziliensis*	PV844901
CF-L03	Dog	Biopsy	Panama Este	N/A	N/A	+	+	*L.* (*V.*) *braziliensis*	PV844903
CF-C04	Dog	Blood	Panama	-	-	-	-	N/A	N/A
CF-A05	Dog	Serum/ Blood	Panama Oeste	-	-	-	-	N/A	N/A
CF-S06	Dog	Serum/ Blood	Panama Oeste	-	-	-	-	N/A	N/A
CF-J07	Dog	Serum/ Blood	Panama	+	+	+	+	N/A	N/A
CF-F08	Dog	Serum/ Blood	Panama	-	-	-	-	N/A	N/A
CF-A09	Dog	Serum/ Blood	Panama Norte	+	+	+	+	N/A	N/A
CF-S10	Dog	Serum/ Blood	Panama Norte	-	-	-	-	N/A	N/A
EC-H01	Horse	Biopsy	Panama Oeste	N/A	N/A	+	+	*L.* (*M.*) *martiniquensis*	PV658270 PV844900
EC-H02	Horse	Biopsy	Panama Este	N/A	N/A	-	-	N/A	N/A

* PCR kDNA includes *Viannia* genus-specific PCR and L150/L151 PCR. +: Positive. -: Negative

## Data Availability

The sequences used to support the findings of this study are available from GenBank under the accession numbers: PV658270 and PV844900-PV844903.

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
