# Peer review of "Molecular Detection of Leishmania (V.) braziliensis and Leishmania (M.) martiniquensis Infecting Domestic Animals from Panama, Central America"

_animals, 2025, doi:10.3390/ani15182677_

Round 1
Reviewer 1 Report
Comments and Suggestions for Authors
The manuscript addresses the Molecular detection of Leishmania (V.) braziliensis and Leishmania (M.) martiniquensis infecting domestic animals from Panama, Central America. Although L. panamensis is the predominant species, L. braziliensis is known to circulate in Panama, in humans and dogs since the seventies, but molecular analysis was not yet been performed, so it is relevant. L. martiniquensis is unknown in Panamá.
May main concerns are the methodology and their results.
Material and methods
In Study design, the specific origin of samples (regions) should be stated.
More details should be given on the serological techniques, in IFA the antigen, cut-off should be referred.
The in PCR-RFLP ITS, it is not perceptible that the methodology in reference 21 was used for L. martiniquensis typing, but rather for Old World leishmania species; more details should be given for better clarification on the L. martiniquensis profile.
More details on the phylogenetic analysis are needed, the authors should specify the steps taken to produce the trees, including replicates for bootstrap and outgroups.
Results
Table 1 should contain a column with animal species (dog, horse).
Instead of origin, maybe region or city; remove PA, as we no already that the samples are from Panama (PA); on the table legend include the meaning of N/A;
Figure one is imperceptible, too small; nevertheless, it seems that bootstrap values presented in the phylogenetic trees, using each analysis one gene sequence, are mostly low; so, the support for the inferred analysis is limited and rises concerns on the conclusions which are taken. I suggest that a more robust analysis be performed combining multiple genes. Moreover, authors should recognise this limitation on the discussion.
Discussion
This section should be reorganised upon revision of results; the existence of L. braziliensis/ L. panamensis hybrids should be discussed; moreover, more info should be given on Leishmania vectors in the country and their role in transmission of L. martiniquensis which may circulate in Panama.
Conclusions
Should be reformulated in the light of the limited analysis in its present form.
Figure S2 bring no added value for the manuscript. It should be removed.
Author Response
Thank you very much for taking the time to review this manuscript. Please find the detailed responses below and the corresponding revisions/corrections highlighted/in track changes in the re-submitted files.
|
2. Point-by-point response to Comments and Suggestions for Authors |
|
Material and methods
Comments 1: In Study design, the specific origin of samples (regions) should be stated.
Response 1: Thank you for your comment. We agree with the observation. Information on the origin (regions) of the samples has been added to the manuscript text, and also added in Table 2 of the revised manuscript.
Line 111-115: “The suspected samples originated from ten dogs (Canis lupus familiaris) and two horses (Equus ferus caballus) from endemic regions of cutaneous leishmaniasis (CL) in the country, within the areas of Panama (Panama City), Panama Oeste (La Chorrera), Panama Este (Chepo), Panama Norte (Chilibre), and Colon (Escobal).”
Comments 2: More details should be given on the serological techniques, in IFA the antigen, cut-off should be referred.
Response 2: Thank you very much for your comment. Information regarding the IFA methodology has been added to the manuscript.
Line 118-128: “Serum samples were analyzed for the detection of Leishmania antibodies using two serological methods. The Indirect Immunofluorescence Assay (IFA) detects anti-Leishmania antibodies by incubating serum samples with antigen-coated slides, followed by binding of a fluorescent secondary antibody and microscopic visualization. The antigen was obtained from L. (V.) panamensis MHOM/BR/71/LS94 promastigotes, and we used an anti-Dog IgG (whole molecule) – FITC antibody produced in rabbit (Sigma: F7884). A titer of ≥1:64 was considered positive, based on prior validation studies in dogs from Panama [18]. The second method was a rapid immunochromatographic test (DPP®, BioManguinhos, Brazil), which detects antibodies against the rK39 recombinant antigen using lateral flow technology and is commonly used for the diagnosis of canine visceral leishmaniasis. Results were interpreted visually according to the manufacturer’s instructions.”
Comments 3: The in PCR-RFLP ITS, it is not perceptible that the methodology in reference 21 was used for L. martiniquensis typing, but rather for Old World leishmania species; more details should be given for better clarification on the L. martiniquensis profile.
Response 3: We thank the reviewer for the important observation. Indeed, the original methodology described by Spanakos et al. (2008) was designed for the detection and typing of Old World Leishmania species using the ITS1 region. However, in our study, we applied this ITS1-based PCR complemented with a sequencing approach to confirm the identity of Leishmania (Mundinia) martiniquensis, which is phylogenetically close to several Old-World species and falls outside the traditional Viannia and Leishmaniasubgenera. Although L. martiniquensis is now increasingly reported worldwide, including in the Americas, there is currently no universally standardized ITS1-RFLP protocol for this species. Therefore, we relied on ITS1 amplification followed by Sanger sequencing and phylogenetic analysis to determine species identity. Additionally, we sequenced a second molecular marker (Hsp70 gene). This dual-marker approach increased the robustness of our molecular identification.
Recent studies conducted by different authors have also used this gene for the description of L. martiniquensis:
- Nopporn Songumpai et al. First Evidence of Co-Circulation of Emerging Leishmania martiniquensis, Leishmania orientalis, and Crithidia sp. in Culicoides Biting Midges (Diptera: Ceratopogonidae), the Putative Vectors for Autochthonous Transmission in Southern Thailand 2022.
- Saranya Srivarasat et al, Case Report: Autochthonous Disseminated Cutaneous, Mucocutaneous, and Visceral Leishmaniasis Caused by Leishmania martiniquensis in a Patient with HIV/AIDS from Northern Thailand and Literature Review, 2022.
- Narissara Jariyapan, Molecular identification of two newly identified human pathogens causing leishmaniasis using PCR-based methods on the 30 untranslated region of the heat shock protein 70 (type I) gene; 2021.
For a better understanding of this point, the following information has been added:
Line 211-218: “In the case of the horse, amplification and Sanger sequencing of the ITS1 region of the rRNA gene (379 bp product) were performed to confirm L. (M.) martiniquensis infection. Although the methodology referenced [24] was originally developed for Old World Leishmania species, it was suitable for L. (M.) martiniquensis due to its phylogenetic placement within the Mundinia subgenus, which clusters near Old World species [22,23,25]. ITS1 sequences were analyzed by BLAST and phylogenetic reconstruction to establish species identity. A second molecular approach (Hsp-70 genes sequencing) was used to confirm this result.”
Comments 4: More details on the phylogenetic analysis are needed, the authors should specify the steps taken to produce the trees, including replicates for bootstrap and outgroups.
Response 4: We appreciate the reviewer’s comment and agree that additional details regarding the phylogenetic analyses are warranted. In response, we have now included a more thorough description of the methods used to construct the phylogenetic trees. Specifically, we clarify the alignment procedures, evolutionary models, number of bootstrap replicates, and the outgroups used. These details have been added to the "Sequencing" subsection (2.4) in the revised manuscript.
Line 227-244: “To create the phylogenetic tree of the Hsp-70 gene with the sequences of L. (V.) braziliensis and L. (M.) martiniquensis, the best-fit model TN+F+I, and an ultra-fast bootstrap with 1,000 replicates were used. In the case of the ITS1 gene with the sequence of L. (M.) martiniquensis, the best-fit model HKY+F+I and an ultra-fast bootstrap with 1,000 repli-cates were employed. Representative sequences obtained from NCBI GenBank of Leishmania and Endotrypanum were used for phylogenetic analysis with the Hsp-70 gene, and existing sequences of Leishmania were used for phylogenetic analysis with the ITS1 gene.”
Results
Comments 5: Table 1 should contain a column with animal species (dog, horse).
Instead of origin, maybe region or city; remove PA, as we no already that the samples are from Panama (PA); on the table legend include the meaning of N/A;
Response 5: We thank the reviewer for these helpful suggestions regarding Table 1. In response, we have added a new column specifying the animal species (dog or horse) for each sample; the column labeled “Origin” has been renamed to “Region” and the country code “PA” has been removed, as all samples originated from Panama. We have also updated the table legend to explain the abbreviation “N/A” as “Not Available” or “Not Applicable,” depending on the context.
By changing the “origin” to “Region” of domestic animals, we were able to obtain more accurate information, which was incorporated into the table.
Table 2. Detection of Leishmania in domestic animals, including diagnostic results, species identification, and region. N/A: Not Available or Not Applicable.
ID |
Animal Species |
Sample Type |
Region |
DPP rk39 |
IFI |
PCR kDNA* |
PCR Hsp-70 |
LeishmaniaSpecies |
GenBank Accession |
CF-W01 |
Dog |
Biopsy |
Colon |
N/A |
N/A |
+ |
+ |
L. (V.) braziliensis |
PV844902 |
CF-P02 |
Dog |
Biopsy |
Colon |
N/A |
N/A |
+ |
+ |
L. (V.) braziliensis |
PV844901 |
CF-L03 |
Dog |
Biopsy |
Panama Este |
N/A |
N/A |
+ |
+ |
L. (V.) braziliensis |
PV844903 |
CF-C04 |
Dog |
Blood |
Panama |
- |
- |
- |
- |
N/A |
N/A |
CF-A05 |
Dog |
Serum/ Blood |
Panama Oeste |
- |
- |
- |
- |
N/A |
N/A |
CF-S06 |
Dog |
Serum/ Blood |
Panama Oeste |
- |
- |
- |
- |
N/A |
N/A |
CF-J07 |
Dog |
Serum/ Blood |
Panama |
+ |
+ |
+ |
+ |
N/A |
N/A |
CF-F08 |
Dog |
Serum/ Blood |
Panama |
- |
- |
- |
- |
N/A |
N/A |
CF-A09 |
Dog |
Serum/ Blood |
Panama Norte |
+ |
+ |
+ |
+ |
N/A |
N/A |
CF-S10 |
Dog |
Serum/ Blood |
Panama Norte |
- |
- |
- |
- |
N/A |
N/A |
EC-H01 |
Horse |
Biopsy |
Panama Oeste |
N/A |
N/A |
+ |
+ |
L. (M.) martiniquensis |
PV658270 PV844900 |
EC-H02 |
Horse |
Biopsy |
Panama Este |
N/A |
N/A |
- |
- |
N/A |
N/A |
*PCR kDNA includes VianniaGenus-Specific PCR and L150/L151 PCR.
Comments 6: Figure one is imperceptible, too small; nevertheless, it seems that bootstrap values presented in the phylogenetic trees, using each analysis one gene sequence, are mostly low; so, the support for the inferred analysis is limited and rises concerns on the conclusions which are taken. I suggest that a more robust analysis be performed, combining multiple genes. Moreover, authors should recognise this limitation on the discussion.
Response 6: We thank the reviewer for this valuable observation. We acknowledge that the bootstrap support values in the phylogenetic trees—particularly in the ITS1 analysis—were limited in some branches. This limitation is likely due to the use of single-locus markers, which generally provide lower phylogenetic resolution compared to multilocus datasets. While the Hsp70 and ITS1 markers independently supported a preliminary and reliable species identification, we agree that a more robust inference could be achieved through multilocus sequence analysis or whole-genome approaches.
In addition to the molecular findings, the clinical and epidemiological context strengthens our identification. The cutaneous lesion characteristics observed in the horse, along with its origin in a tropical region, are consistent with previously reported cases of leishmaniasis caused by Leishmania martiniquensis. Taken together, the molecular, clinical, and epidemiological evidence strongly supports L. martiniquensis as the etiological agent in this case.
To address this observation, we have acknowledged this as a limitation in the Discussion section of the revised manuscript. It is important to mention that our conclusions regarding species identity are supported by both sequence homology (BLAST) and concordant results from two markers (Hsp70 and ITS1).
We have also increased the resolution of Figure 1 for better readability and visualization of bootstrap values. We plan to pursue deeper phylogenomic analyses in future work, as we expand our molecular surveillance of Leishmania species in domestic and wild hosts.
Line 432-445: “One limitation of our study is the use of single-locus markers (Hsp70 and ITS1) for phylogenetic reconstruction, which may lead to lower bootstrap support in certain branches of the trees, as observed in Figure 1. While these genes are commonly used for Leishmania species identification [20,34,38,39], they may not provide sufficient resolution for deeper evolutionary relationships. The use of multilocus or whole-genome sequencing approaches would improve phylogenetic robustness and should be considered in future studies. Nevertheless, the consistent results across two genetic markers and the high sequence identity with reference strains support the accuracy of our species-level identifications. In addition to the molecular findings, the clinical and epidemiological context strengthens our identification. The cutaneous lesion characteristics observed in the horse, along with its origin in a tropical region, are consistent with previously reported cases of leishmaniasis caused by L. (M.) martiniquensis [34,35]. Taken together, the molecular, clinical, and epidemiological evidence strongly supports L. (M.) martiniquensis as the etiological agent in this case.”
Discussion
Comments 7: This section should be reorganised upon revision of results; the existence of L. braziliensis/ L. panamensis hybrids should be discussed; moreover, more info should be given on Leishmania vectors in the country and their role in transmission of L. martiniquensis which may circulate in Panama.
Response 7: We appreciate the reviewer’s important suggestions. In response: We have revised and reorganized the Discussion section to better align with the results presented; We have added a paragraph addressing the potential existence of Leishmania (Viannia) braziliensis/panamensis hybrids, as previously reported in Panama, and their relevance in interpreting species identification based on molecular markers. Additionally, we now include more information on sand fly vectors in Panama, including their known associations with L. (V.) panamensis, L. (V.) braziliensis, and potential roles in the transmission of L. (M.) martiniquensis, whose vector remains unidentified in the region. These additions provide better context to our findings and underscore the need for continued entomological and molecular surveillance in Panama.
Line 393-406: “It is also important to consider the possibility of Leishmania (Viannia) hybrid strains in Panama. A previous study conducted in Panama detected molecular signatures suggestive of L. (V.) braziliensis/L. (V.) panamensis hybrids in human cutaneous lesions [4]. Although phylogenetic analysis demonstrated the presence of L. (V.) braziliensis / L. (V.) guyanensis hybrids, we cannot rule out the possibility of the presence of both hybrids in the country. Such hybrids may complicate molecular identification, particularly when using single-locus markers such as Hsp-70. Although our Hsp70-based phylogeny grouped the sequences within the L. (V.) braziliensis clade, we cannot rule out the presence of genetic introgression or hybridization with L. (V.) panamensis, especially given the geographic overlap of these species in endemic areas. Future studies using multilocus genotyping or whole-genome sequencing will be essential to clarify this issue.”
Line 414-423: “In Panama, several species of Lutzomyia sand flies have been implicated in the transmission of L. (V.) panamensis and L. (V.) braziliensis, particularly Lutzomyia gomezi, L. panamensis, and L. trapidoi [7,35,36] However, the vector responsible for the transmission of L. (M.) martiniquensis remains unknown. In other endemic areas such as Southeast Asia, Sergentomyia spp. and Phlebotomus spp. have been suggested as potential vectors of Mundinia species [33]. Given the presence of L. martiniquensis in a horse from Panama without travel history, and its phylogenetic confirmation, it is plausible that local phlebotomine species may be involved in its transmission. Entomological investigations are needed to identify potential vector species and assess their infection rates with Mundinia parasites in Central America.”
Conclusions
Comments 8: Should be reformulated in the light of the limited analysis in its present form.
Response 8: We acknowledge the reviewer’s concern regarding the limited scope of the current molecular analysis and agree that the conclusions should reflect this constraint. We have revised the Conclusion section to provide a more cautious and balanced interpretation of our findings, explicitly acknowledging the limitations of single-locus phylogenetic inference and highlighting the need for future studies using multilocus or genomic approaches.
Line 463-472: “Our findings provide preliminary molecular evidence of the diversity of Leishmania species infecting domestic animals in Panama, including the first documentation of L. (V.) braziliensis in local dogs and L. (M.) martiniquensis in a horse. While these results expand our understanding of potential animal involvement in the transmission cycle of cutaneous leishmaniasis, the analyses were based on two single locus markers, which generally provided lower phylogenetic resolution. Consequently, species identification should be interpreted with caution. Future studies using multilocus genotyping or genomic approaches are essential to confirm these findings and to explore the possibility of hybrid strains or cryptic diversity. Nonetheless, our results underscore the importance of strengthening surveillance efforts in domestic animals under a One Health framework to better understand the eco-epidemiology of Leishmania transmission in Central America.”
Comments 9: Figure S2 bring no added value for the manuscript. It should be removed.
Response 9: We thank the reviewer for this observation. We agree that Figure S2 does not substantially enhance the scientific value of the manuscript and have removed it accordingly.

Reviewer 2 Report
Comments and Suggestions for Authors
It would be useful to indicate in the abstract the total number of animals included in the study, and eventually the percentage of positive animals for leishmaniasis.
Line 31, it would be better to indicate ‘sandflies’ directly instead of ‘insects’, given that the parasite is classically transmitted by sandflies.
Line 33: a dot is missing
Line 41, perhaps write ‘it is essential’ instead of ‘it's a good idea’
Lines 73 and 195, check that reference 3 is the correct bibliographical reference
Line 83, instead of reference 14, a more general article on canine leishmaniasis in Brazil could be cited.
In materials and methods :
- for section ‘2.2 Serological Analysis’, it would be better to briefly describe the technique.
- for the ‘2.3.2 PCR’ section, it would be better to give the sequences of the primers used in a table, and eventually the PCR conditions such as primer concentrations and PCR cycles.
On line 120, it could go further than saying that these primers target ‘kDNA Viannia Genus’, by saying that these primers target the ‘L. braziliensis complex’
On lines 127-128, information related to the PCR could be placed in the ‘2.3.2 PCR’ section, with all the information on primer sequences and PCR conditions
On lines 138-139, check the accession numbers PV844900-PV844903 because they are not accessible on the GenBank site. However, PV658270 is accessible.
Table 1 lacks 1 horse.
In the legend of Table 1, the abbreviations N/A could be stipulated.
Line 161-162: it is stated that the remaining dogs were tested for suspected CVL. It would be useful to indicate the symptoms of these dogs.
Similarly, line 190-191, it would be useful to indicate the symptoms of the other horse.
Line 192, perhaps indicate very briefly what this disease is.
Line 213, bibliographic reference 25 does not seem to correspond to the indicated text.
Author Response
Thank you very much for taking the time to review this manuscript. Please find the detailed responses below and the corresponding revisions/corrections highlighted/in track changes in the re-submitted files.
|
2. Point-by-point response to Comments and Suggestions for Authors |
|
Comments 1: It would be useful to indicate in the abstract the total number of animals included in the study, and eventually the percentage of positive animals for leishmaniasis.
Response 1: We appreciate this helpful suggestion. As recommended, we have revised the Abstract to include the total number of animals analyzed (n=12) and the proportion of animals that tested positive for Leishmania infection (6/12; 50%).
Line 33-35: In this study, we evaluated twelve domestic animals with suspected leishmaniasis lesions between 2021 and 2025. Six of them (50%) tested positive: five dogs and one horse.
Line 53-56: “In this study, samples from twelve domestic animals (ten dogs and two horses) with suspected CL lesions were collected between 2021 and 2025 in endemic regions of Panama and evaluated using multiple diagnostic methods. Leishmania infection was confirmed in six of them (50%): five dogs and one horse.”
Comments 2: Line 31, it would be better to indicate ‘sandflies’ directly instead of ‘insects’, given that the parasite is classically transmitted by sandflies.
Response 2: Thank you for this valuable suggestion. We agree that specifying "sandflies" instead of the more general term "insects" improves the accuracy of the Simple Summary. The text has been revised accordingly.
Line 31: Cutaneous leishmaniasis is a disease transmitted by sandflies that mainly affects the skin.
Comments 3: Line 33: a dot is missing
Response 3: Thanks for the comment. The point has been added in the text.
Comments 4: Line 41, perhaps write ‘it is essential’ instead of ‘it's a good idea’
Response 4: Thanks for the comment. "it’s good idea" has been replaced with "it’s essential."
Line 41-42: So, it’s essential to keep an eye on both people and animals, particularly in rural areas,
Comments 5: Lines 73 and 195, check that reference 3 is the correct bibliographical reference
Response 5: The reference “Pan American Health Organization Leishmaniasis: Epidemiological Report for the Americas; Washington, D.C., 2024”, describes the Leishmania species circulating in the different countries of the Americas. This reference was included because of the general and up-to-date information it provides in the context of the information presented in the manuscript. IN the revised manuscript we have added another review article by Lainson, which describes the distribution of Leishmania species from the Neotropics (America). “Lainson, R. The Neotropical Leishmania Species: A Brief Historical Review of Their Discovery, Ecology and Taxonomy. Rev Panamazonica Saude 2010”.
Comments 6: Line 83, instead of reference 14, a more general article on canine leishmaniasis in Brazil could be cited.
Response 6: Thank you, we agree with this comment. In response, we've considered including a representative reference based on the text. In response to your comment, we've included two review articles on the description of Leishmania in wild and domestic animals in Brazil and removed reference 14.
- A systematic review and meta-analysis of the factors associated with Leishmania infantum infection in dogs in Brazil. Belo V, Struchiner C et al.Veterinary Parasitology (2013)
- Identification of infection by Leishmania spp. in wild and domestic animals in Brazil: a systematic review with meta-analysis (2001–2021). Ratzlaff F, Osmari V et al. Parasitology Research (2023).
In materials and methods :
Comments 7: - for section ‘2.2 Serological Analysis’, it would be better to briefly describe the technique.
Response 7: Thank you for your feedback. For a better understanding, to clarify this point, information on serological methodologies has been added in the revised manuscript.
Line 118-128: “Serum samples were analyzed for the detection of Leishmania antibodies using two serological methods. The Indirect Immunofluorescence Assay (IFA) detects anti-Leishmania antibodies by incubating serum samples with antigen-coated slides, followed by binding of a fluorescent secondary antibody and microscopic visualization. The antigen was obtained from L. (V.) panamensis promastigotes, and we used an anti-Dog IgG (whole molecule) – FITC antibody produced in rabbit (Sigma: F7884). A titer of ≥1:64 was considered positive, based on prior validation studies in dogs from Panama [18]. The second method was a rapid immunochromatographic test (DPP®, BioManguinhos, Brazil), which detects antibodies against the rK39 recombinant antigen using lateral flow technology and is commonly used for the diagnosis of canine visceral leishmaniasis. Results were interpreted visually according to the manufacturer’s instructions.”
Comments 8: - for the ‘2.3.2 PCR’ section, it would be better to give the sequences of the primers used in a table, and eventually the PCR conditions such as primer concentrations and PCR cycles.
Response 8: Thank you very much for your comment. Following your suggestion, we have added a table with the requested information on the PCRs used in the study.
Table 1. Description of the primers and conditions of the PCRs used to identify Leishmania parasites from domestic animals.
PCR |
Primers |
Primers concentrations |
DNA used per reaction |
PCR conditions |
PCR products size |
kDNA Viannia |
B1: 5′-GGGGTTGGTGTAATATAGTGG-3′ LV: 5′-ATTTTTGAACGGGGTTTCTG-3′
|
0.6mm/L each one
|
5mL |
5 cycles of 95°C x 6 min 95°C x 30 seg 64.5°C x 2 min 72°C x 1 min 35 cycles of 95°C x 30 seg 64°C x 1 min 72°C x 1 min 72°C x 10 min |
750 bp |
Hsp-70 |
Hsp70-F25: 5’-GGACGCCGGCACGATTKCT-3’ Hsp70-R1310: 5’-CCTGGTTGTTGTTCAGCCACTC-3’ |
0.6mm/L each one
|
5mL |
94°C x 5 min 33 cycles of 94°C x 30 seg 61°C x 1 min 72°C x 3 min 72°C x 10 min |
1286 bp |
kDNA L150/151 |
L-150: 5’-GGG(G/T)AGGGGCGTTCT(G/C)CGAA-3’ L-151: 5’-(G/C)(G/C)(G/C)A/C)CTAT(A/T)TTACACCAACCCC-3’ |
0.5mm/L each one
|
5mL |
94°C x 4 min 33 cycles of 94°C x 30 seg 52.3°C x 30 seg 72°C x 30 seg 72°C x 10 min |
120 bp |
ITS1 |
LeR: 5’-CCAAGTCATCCATCGCGACACG-3’ LeF 5‘-TCCGCCCGAAAGTTCACCGATA-3’) |
1.0mm/L each one
|
5mL |
95°C x 5 min 40 cycles of 95°C x 1 min 65°C x 1 min 72°C x 1 min 72°C x 10 min |
379 bp |
Comments 9:On line 120, it could go further than saying that these primers target ‘kDNA Viannia Genus’, by saying that these primers target the ‘L. braziliensis complex’
Response 9:Thank you for your comment. As suggested, this information has been added for better understanding.
Line 137-139: “Four conventional PCR tests were performed for the detection of Leishmania (Table 1): a) kDNA VianniaGenus-Specific PCR: targeting the kinetoplast minicircle region for the detection of LeishmaniaViannia subgenus including L. braziliensis complex [19];”
Comments 10: On lines 127-128, information related to the PCR could be placed in the ‘2.3.2 PCR’ section, with all the information on primer sequences and PCR conditions
Response 10: Thank you for your comment. Based on “Comment 8” the information requested has been added in Table 1 for better visualization, including the primer sequence, concentrations used, amount of DNA per reaction, PCR conditions, and amplified product.
Comments 11: On lines 138-139, check the accession numbers PV844900-PV844903 because they are not accessible on the GenBank site. However, PV658270 is accessible.
Response 11: Thank you very much for your comment. We reviewed the access codes for the sequences deposited in GenBank, and we have recently verified that all sequences are accessible. They were reviewed on July 22, 2025. It is possible that some sequences were unavailable at the time you try to access GenBank. We are sorry for the inconvenience.
Comments 12: Table 1 lacks 1 horse.
In the legend of Table 1, the abbreviations N/A could be stipulated.
Response 12: Thank you very much for the comment. Indeed, information about one horse was missing. It was added to the revised table, and the meaning of the abbreviation N/A was added.
Table 2. Detection of Leishmania in domestic animals, including diagnostic results, species identification, and region. N/A: Not available or Not Applicable.
ID |
Animal Species |
Sample Type |
Region |
DPP rk39 |
IFI |
PCR kDNA* |
PCR Hsp-70 |
LeishmaniaSpecies |
GenBank Accession |
CF-W01 |
Dog |
Biopsy |
Colon |
N/A |
N/A |
+ |
+ |
L. (V.) braziliensis |
PV844902 |
CF-P02 |
Dog |
Biopsy |
Colon |
N/A |
N/A |
+ |
+ |
L. (V.) braziliensis |
PV844901 |
CF-L03 |
Dog |
Biopsy |
Panama Este |
N/A |
N/A |
+ |
+ |
L. (V.) braziliensis |
PV844903 |
CF-C04 |
Dog |
Blood |
Panama |
- |
- |
- |
- |
N/A |
N/A |
CF-A05 |
Dog |
Serum/ Blood |
Panama Oeste |
- |
- |
- |
- |
N/A |
N/A |
CF-S06 |
Dog |
Serum/ Blood |
Panama Oeste |
- |
- |
- |
- |
N/A |
N/A |
CF-J07 |
Dog |
Serum/ Blood |
Panama |
+ |
+ |
+ |
+ |
N/A |
N/A |
CF-F08 |
Dog |
Serum/ Blood |
Panama |
- |
- |
- |
- |
N/A |
N/A |
CF-A09 |
Dog |
Serum/ Blood |
Panama Norte |
+ |
+ |
+ |
+ |
N/A |
N/A |
CF-S10 |
Dog |
Serum/ Blood |
Panama Norte |
- |
- |
- |
- |
N/A |
N/A |
EC-H01 |
Horse |
Biopsy |
Panama Oeste |
N/A |
N/A |
+ |
+ |
L. (M.) martiniquensis |
PV658270 PV844900 |
EC-H02 |
Horse |
Biopsy |
Panama Este |
N/A |
N/A |
- |
- |
N/A |
N/A |
Comments 13: Line 161-162: it is stated that the remaining dogs were tested for suspected CVL. It would be useful to indicate the symptoms of these dogs.
Similarly, line 190-191, it would be useful to indicate the symptoms of the other horse.
Line 192, perhaps indicate very briefly what this disease is.
Response 13: Thank you for your feedback. Clinical information has been added on the remaining five dogs with suspected CVL and the horse diagnosed with habronemiasis. A brief explanation of habronemiasis has been provided.
Line 291-293: The remaining five dogs with suspected CVL presented non-specific symptoms, including weight loss, lethargy, enlarged lymph nodes, and different types of skin lesions.
Line 352-358: The CL-negative horse had an ulcerated lesion with crust, purulent discharge, and intense itching. This horse was diagnosed with cutaneous habronemiasis, a parasitic disease of horses frequently prevalent in tropical climates caused by Habronema larvae transmitted by flies (Musca domestica and Stomoxys calcitrans, which cause skin, eye, or gastric lesions. Gross lesions are manifest as granulomatous ulcers that can be confused with ulcerated lesions caused by Leishmania parasites [27,28].
Comments 14: Line 213, bibliographic reference 25 does not seem to correspond to the indicated text.
Response 14: Thank you for the comment. The reviewer is correct: the reference was in the wrong place.
It was placed in the following paragraph:
Line 377-378: In South America, L. (V.) braziliensis is the most common causative agent of ACL in dogs [10,32].

Reviewer 3 Report
Comments and Suggestions for Authors
In this study, authors describe the presence of DNA of Leishmania in dogs and a horse. Samples from domestic animals with CL lesions were confirmed as leishmaniasis by immunological and molecular methods. CL was confirmed in six animals: five dogs and one horse. Dogs were infected with Leishmania (Viannia) braziliensis, and Leishmania (Leishmania) infantum and the horse was infected with L. (M.) martiniquensis.
The manuscript is well written and clear and some minor points must be clarified:
1) Describe in methods how ITS1 was amplified. Please include the reference.
2) Include the amount of DNA used per PCR reaction. This information is missiing.
3) Lines 168-169: Authors said that for two CVL cases, Hsp70-RFLP and sequencing were not possible due to insufficient amplified DNA. Did you check the quality of the DNA? Did you try to amplify the genomic DNA of the host?
Author Response
Thank you very much for taking the time to review this manuscript. Please find the detailed responses below and the corresponding revisions/corrections highlighted/in track changes in the re-submitted files.
|
2. Point-by-point response to Comments and Suggestions for Authors |
|
Comments 1:
1) Describe in methods how ITS1 was amplified. Please include the reference.
2) Include the amount of DNA used per PCR reaction. This information is missing.
Response 1: Thank you very much for your comment. Information on the PCRs used in the study has been added to a table based on your observations and those of the other reviewers.
Table 1. Description of the primers and conditions of the PCRs used to identify Leishmania parasites from domestic animals.
PCR |
Primers |
Primers concentrations |
DNA used per reaction |
PCR conditions |
PCR products size |
kDNA Viannia |
B1: 5′-GGGGTTGGTGTAATATAGTGG-3′ LV: 5′-ATTTTTGAACGGGGTTTCTG-3′
|
0.6mm/L each one
|
5mL |
5 cycles of 95°C x 6 min 95°C x 30 seg 64.5°C x 2 min 72°C x 1 min 35 cycles of 95°C x 30 seg 64°C x 1 min 72°C x 1 min 72°C x 10 min |
750 bp |
Hsp-70 |
Hsp70-F25: 5’-GGACGCCGGCACGATTKCT-3’ Hsp70-R1310: 5’-CCTGGTTGTTGTTCAGCCACTC-3’ |
0.6mm/L each one
|
5mL |
94°C x 5 min 33 cycles of 94°C x 30 seg 61°C x 1 min 72°C x 3 min 72°C x 10 min |
1286 bp |
kDNA L150/151 |
L-150: 5’-GGG(G/T)AGGGGCGTTCT(G/C)CGAA-3’ L-151: 5’-(G/C)(G/C)(G/C)A/C)CTAT(A/T)TTACACCAACCCC-3’ |
0.5mm/L each one
|
5mL |
94°C x 4 min 33 cycles of 94°C x 30 seg 52.3°C x 30 seg 72°C x 30 seg 72°C x 10 min |
120 bp |
ITS1 |
LeR: 5’-CCAAGTCATCCATCGCGACACG-3’ LeF 5‘-TCCGCCCGAAAGTTCACCGATA-3’) |
1.0mm/L each one
|
5mL |
95°C x 5 min 40 cycles of 95°C x 1 min 65°C x 1 min 72°C x 1 min 72°C x 10 min |
379 bp |
Comments 3: 3) Lines 168-169: Authors said that for two CVL cases, Hsp70-RFLP and sequencing were not possible due to insufficient amplified DNA. Did you check the quality of the DNA? Did you try to amplify the genomic DNA of the host?
Response 3:Thank you very much for this observation. We assessed the DNA concentration and purity of both samples using the NanoDrop Lite Plus Spectrophotometer (ThermoFisher Scientific), obtaining concentrations of 45.3 ng/µL and 99.4 ng/µL, respectively. However, despite seemingly adequate concentrations, we believe that DNA quality was suboptimal, as evidenced by the low intensity of the amplified bands observed in agarose gels. When subjected to sequencing, the resulting sequences showed numerous gaps and low-quality signals, which prevented the generation of reliable sequences and their inclusion in the phylogenetic analysis. We suspect that partial DNA degradation and/or the presence of PCR inhibitors may have negatively affected both amplification efficiency and sequencing quality, despite acceptable initial concentration readings. We attempted to repeat the PCR under optimized conditions; however, amplification/sequencing remained unsuccessful. Unfortunately, we were unable to obtain additional biological material from these cases for further analysis.

Round 2
Reviewer 1 Report
Comments and Suggestions for Authors
I appreciate the alterations made on the manuscript which improved its quality.
There are just two minor changes that authors my include:
- Simplify the Table1 title, as the table itself is auto-explanative, to “PCR protocols used to identify Leishmania parasites from domestic animals”
- Include the reference of each PCR protocol in table 1, either within the table or as footnote.
Author Response
Thank you very much for taking the time to review this manuscript. Your comments have helped improve the manuscript. Please find the detailed responses below and the corresponding revisions/corrections highlighted/in track changes in the re-submitted files.
Point-by-point response to Comments and Suggestions for Authors |
|
There are just two minor changes that authors my include:
Comments 1: - Simplify the Table1 title, as the table itself is auto-explanative, to “PCR protocols used to identify Leishmania parasites from domestic animals”
Comments 2: - Include the reference of each PCR protocol in table 1, either within the table or as footnote.
Response 1 and 2: Thank you for your comments. We concur with the observations. In response to your feedback, we have simplified the title of Table 1 as suggested and included the references to the protocols for each PCR mentioned in Table 1.
Table 1. PCR protocols used to identify Leishmania parasites from domestic animals.
PCR |
Primers |
Primers concentrations |
DNA used per reaction |
PCR conditions |
PCR products size |
kDNA Viannia Vergel et al. 2005 [20]
|
B1: 5′-GGGGTTGGTGTAATATAGTGG-3′ LV: 5′-ATTTTTGAACGGGGTTTCTG-3′
|
0.6mm/L each one
|
5mL |
5 cycles of 95°C x 6 min 95°C x 30 seg 64.5°C x 2 min 72°C x 1 min 35 cycles of 95°C x 30 seg 64°C x 1 min 72°C x 1 min 72°C x 10 min |
750 bp |
Hsp-70 Montalvo et al. 2012 [21] |
Hsp70-F25: 5’-GGACGCCGGCACGATTKCT-3’ Hsp70-R1310: 5’-CCTGGTTGTTGTTCAGCCACTC-3’ |
0.6mm/L each one
|
5mL |
94°C x 5 min 33 cycles of 94°C x 30 seg 61°C x 1 min 72°C x 3 min 72°C x 10 min |
1286 bp |
kDNA L150/151 Marques et al. 2006 [22] |
L-150: 5’-GGG(G/T)AGGGGCGTTCT(G/C)CGAA-3’ L-151: 5’-(G/C)(G/C)(G/C)A/C)CTAT(A/T)TTACACCAACCCC-3’ |
0.5mm/L each one
|
5mL |
94°C x 4 min 33 cycles of 94°C x 30 seg 52.3°C x 30 seg 72°C x 30 seg 72°C x 10 min |
120 bp |
ITS1 Spanakos et al. 2008[25] |
LeR: 5’-CCAAGTCATCCATCGCGACACG-3’ LeF 5‘-TCCGCCCGAAAGTTCACCGATA-3’) |
1.0mm/L each one
|
5mL |
95°C x 5 min 40 cycles of 95°C x 1 min 65°C x 1 min 72°C x 1 min 72°C x 10 min |
379 bp |
